# Resonant Energy Carrier Base Active Charge-Balancing Algorithm

**Mohammad Kamrul Hasan** [1,*], **AKM Ahasan Habib** [2,*], **Shayla Islam** [3],
**Ahmad Tarmizi Abdul Ghani** [1,*] **and Eklas Hossain** [4]

1   Center for Cyber Security, Faculty of Information Science and Technology, University Kebangsaan,
    Bangi 43600, Malaysia
2   Department of Electrical and Computer Engineering, North Garth Institute of Technology,
    Dhaka 1212, Bangladesh
3   Department of Computer Science, Institute of Computer Science and Digital Innovation, UCSI University,
    Kuala Lumpur 56000, Malaysia; shayla@ucsiuniversity.edu.my
4   Department of Electrical and Electronics Engineering, Oregon Institute of Technology,
    Klamath Falls, OR 97601, USA; eklas.hossain@oit.edu
*   Correspondence: mkhasan@ukm.edu.my (M.K.H.); ahasan.diu.eee@gmail.com (A.A.H.);
    atag@ukm.edu.my (A.T.A.G.)

**Abstract:** This paper presents a single LC tank base cell-to-cell active voltage balancing algorithm for Li-ion batteries in electric vehicle (EV) applications. EV batteries face challenges in accomplishing fast balancing and high balancing efficiency with low circuit and control complexity. It addresses that LC resonant tank uses an energy carrier to transfer the voltage from an excessive voltage cell to the lowest voltage cell. The method requires 2N - 4 bidirectional MOSFET switches and a single LC resonant circuit, where N is the number of cells in the battery strings. The balancing speed is improved by allowing a short balancing path for voltage transfer and guarantees a fast balancing speed between any two cells in the battery string, and power consumption is reduced by operating all switches in zero-current switching conditions. The circuit was tested for 4400 mAh Li-ion battery cells under static, cyclic, and dynamic charging/discharging conditions. Two battery cells at the voltage 3.93 V and 3.65 V were balanced after 76 min, and the balancing efficiency is 94.8%. The result of dynamic and cyclic charging/discharging conditions shows that the balancing circuit is applicable for the energy storage devices and Li-ion battery cells for EV.

**Keywords:** active cell balancing; voltage balancing algorithm; Li-ion battery; LC tank; electric vehicle

## 1. Introduction

Day by day, demand for electric vehicles (EV) is increasing compared to internal combustion engine vehicles because of shortage of fossil fuel, environmental problems, and cost-effectiveness. In an internal combustion engine (ICE), when fossil fuel is used as the power source, it produces carbon dioxide ($CO2$) and carbon monoxide (CO), which are harmful to the environment. Besides, fossil fuel could run out in the future. Therefore, the demand for EV increases because pure EV can solve this problem [1,2]. The EV/HEV (hybrid electic vehicle) is regarded as a structure with higher engine efficiency, decreasing the radiation of greenhouse gasses, fuel refilling, or fuel evaporation, and other contaminants that are known as zero-emission vehicles.

EV's critical characteristic is that electric power motor efficiently uses electrical energy from the battery [2,3]. Li-ion batteries are a popular choice because of their quality features [4]; nowadays, super-capacitors also use alternative energy storage. Li-ion battery has high power density, high energy

density, capacity, long lifetime, low self-discharge, memory effect, and low-temperature effect [5,6]. As voltage is very low in a single-cell, so to meet the required load voltage cells are usually connected in the series string. Because of different manufacturing, application, and environmental conditions, however, Li-ion battery has some drawbacks in the battery pack. Overcharged cell in the battery pack has a risk of catching fire, the undercharged cell in the battery pack has a reduced life cycle, and unbalanced charge and discharge gradually reduce the charge capacity and efficiency [5,7–9].

A battery management system (BMS) is essential to improving the battery performance, efficiency, life cycle, capacity, and safety of the battery charge controller. Thus, the BMS is required in performing battery charge balancing. Several researchers worked on BMS as they could improve the battery voltage /charge balancing methods. The battery balancing methods can be separated into two categories: the passive balancing methods and the active balancing methods [10–12]. Authors in References [2,13] classified BMS into two groups, and active balancing has been divided into four sub-groups as shown in Figure 1.

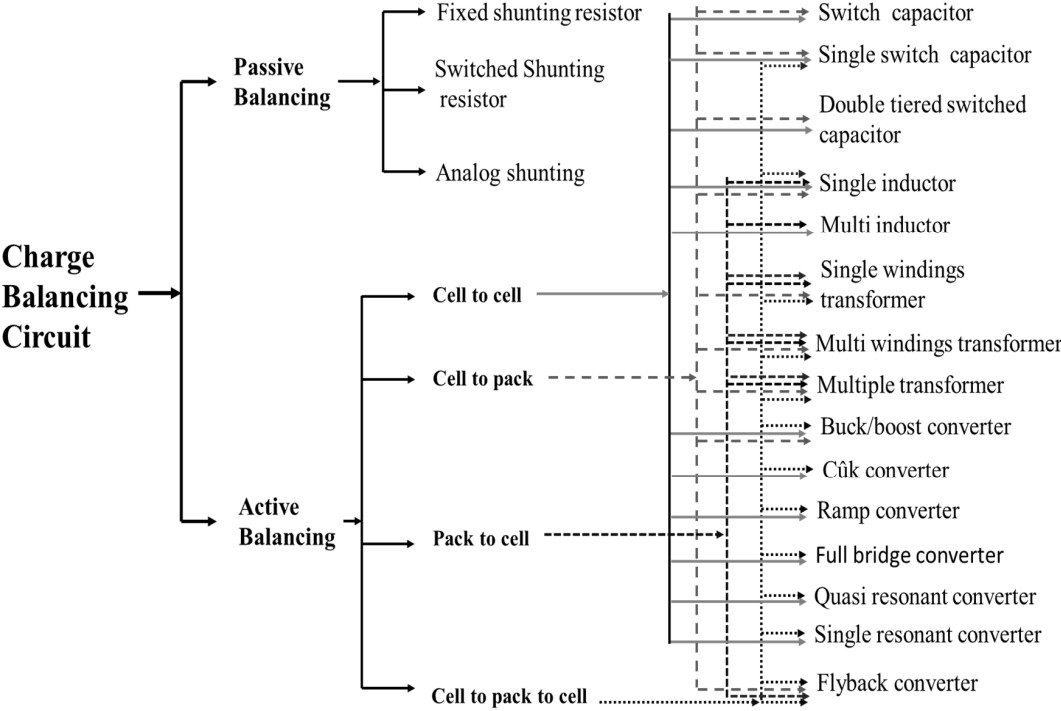

**Figure 1.** Cell balancing topologies.

In passive balancing methods resistor is used for voltage balancing. The excess energy from the higher cell is dissipated by the balancing resistor in the form of heat and balanced with the shortage cell. In active balancing methods, the excess energy from the higher cell is transferred from higher cells to lower cells by the inductor, capacitor, and transistor [14–16]. The pack-to-pack (P2P) [15,17–19] balancing circuits could be efficiently transferred from higher energy storage pack to a lower energy storage pack and take a long balancing time. The cell-to-pack (C2P) [15,20–22] balancing circuits can efficiently discharge the strong cells and inefficiently charge the weak cells. The pack-to-cell (P2C) [15,23–25] balancing circuit can efficiently charge the weak cells and inefficiently discharge the strong cell. The cell-to-cell (C2C) [15,26–44] balancing circuit can charge the weak cells and discharge the healthy cells in the charging, relaxation, and discharging period efficiently. In C2C balancing circuit energy transfer occurs from higher cell to lower cell by single capacitor, inductor, transformer [26–29], multi capacitor, inductor, transformer [30–32], cuk converter [15,33–35], boost converter [35,36], buck-boost converter [34,37–39], ramp converter [15,40], resonant converter [41,42], flyback converter [43–46]. The discussion of all C2C balancing circuits are described in Appendix A.

In terms of execution and control of the switch capacitor base, balancing methods are very simple. In these balancing methods energy recovery takes place when excess energy is transferred from a higher energy storage cell to a lower energy storage cell. This balancing method is imperfect because of switching voltage drops in full balancing time. However, the inductor/transformer-based voltage-balancing method is complicated and effective. This method involves the switching of the excess energy, i.e., the storage cell is turned on, and is connected with the inductor/transformer winding, and transferred to the weak cell. This method has a higher voltage and current stress, magnetizing loss, and implementation problem. On the other hand, converter base voltage-balancing methods are more effective for EV battery voltage balancing. When excess energy transfers from the higher energy storage cells to lower energy storage cells these balancing methods can recover energy, show better efficiency, high equalization speed, and reliability because of bidirectional converter and switches. However when more than two cells are connected in these converters then the control system of these converters become complex, larger in size, and costly. Hence, BMS still requires more research and development in battery voltage balancing for application in EV energy storage systems.

Among the active-balancing system, the C2C balancing process is reliable and most suitable for balancing the system. The C2C balancing system transfers charge to the adjacent C2C and direct C2C in the string through a capacitor, inductor, and transformer. In a direct C2C balancing system, a single capacitor, inductor, and transformer are used to transfer the energy from higher energy capacitive cells to lower energy capacitive cells as shown in Figure 2. The direct C2C single balancing circuits are bidirectional, work on battery cells charging, relaxation mode, and discharge. All of the circuits show low voltage and current stress, good efficiency, miniature size, low cost, are most suitable for low power application, and can be used in P2P or module to module balancing system. However, single C2C balancing circuits are required for complex control systems, sometimes face ripple current.

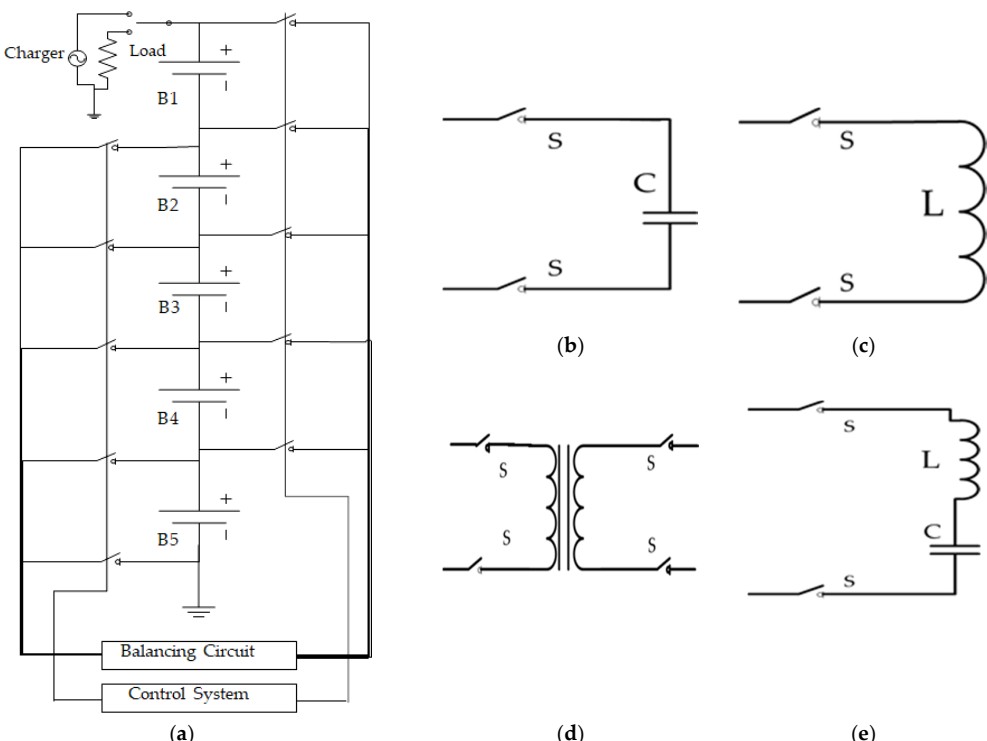

**Figure 2.** Direct C2C balancing system (**a**) basic circuit structure, (**b**) single capacitor-based direct C2C, (**c**) single indictor-based direct C2C, (**d**) single transformer-based C2C, and (**e**) single resonant-converter-based C2C.

This paper presents a single resonant converter based on a first balancing circuit between two cells for different types of electrochemical batteries and balancing algorithms. In this developing circuit,

we reduce the number of bi-directional MOSFET switches and associated components. This paper is organized as follows: Section 2 presents the circuit configuration and the principle of operation; Section 3 presents the voltage-balancing algorithm; Section 4 presents the simulation result of the voltage-balancing circuit; Experiment results and discussion are described in Section 5. Section 6 presents the conclusions.

## 2. Proposed Circuit Configuration and Principle of Operation

### 2.1. Basic Circuit Structure

The proposed voltage-balancing circuit (Figure 3; Figure 3a depicts the proposed circuit, and Figure 3b shows the switch architecture) is a single series LC resonant tank, and all associate switches are connected with a bus line. All cells are series-connected, where MS is a single MOSFET switch, and MD is the bidirectional MOSFET switch. In this circuit, bidirectional MOSFET switches are used to reduce the conduction loss, and all switches are controlled by pulse-width modulation (PWM) single.

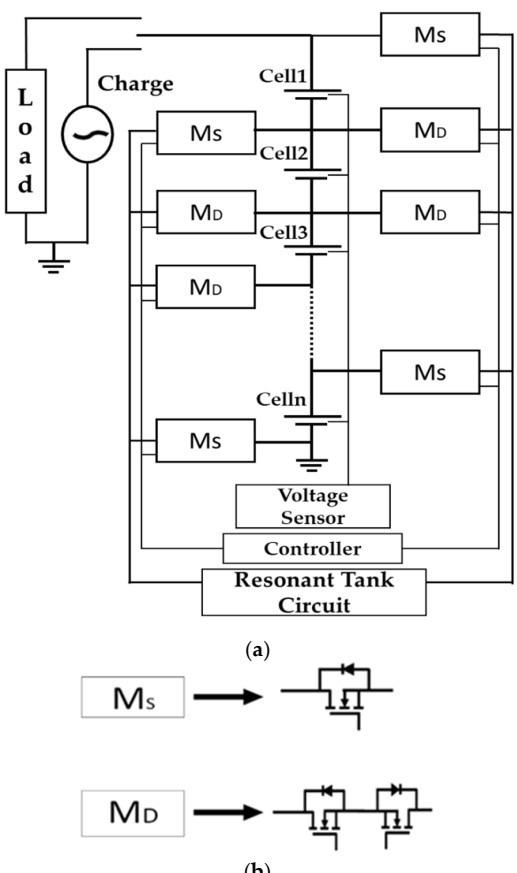

(a)

(b)

**Figure 3.** Proposed single resonant converter-based cell-balancing circuit. (**a**) circutit structure; (**b**) switch configaretion.

The LC circuit stores the higher voltage cell's energy and releases this energy to the lower voltage cell. This proposed circuit has some advantages:

(a). This circuit can balance energy from any battery cell-to-cell.
(b). Reduce the balancing time.
(c). As a single LC tank is used, this circuit becomes small and easy to implement.
(d). All switches are operated in zero voltage condition, so this circuit has less power loss.

## 2.2. Operation Principle

The proposed voltage-balancing circuit's operation principle is understood under the following assumptions (Figures 4 and 5). The excess stored energy is transferred from a higher energy storage cell to a lower energy storage cell by using an LC tank. The cell voltage's operating mode is constant during the balancing operation, and each operation has two types of working modes of charging and discharging. All MOSFET switches are operated by PWM pulse.

### 2.2.1. Working Mode I: Cell1 > Cell2

Charging state: MOSFET switches—$M_S$ is turned ON, and $M_D$ is turned OFF. Cell1, LC tank, and $M_S$ form a clockwise loop in Figure 4a and the excess energy from Cell1 is stored in the LC tank.

Discharge state: MOSFET switches—$M_S$ is turned OFF, and $M_D$ is turn ON. Cell2, LC tank, and $M_D$ form a anti-clockwise loop (as shown in Figure 4b) and LC tank releases the energy stored in Cell2.

### 2.2.2. Working Mode II: Cell2 > Cell1

Charging state: MOSFET switches—$M_D$ is turned ON and $M_S$ is turned OFF. Cell2, LC tank, and $M_D$ form a clockwise loop in Figure 4c and the excess energy from Cell2 is stored in the LC tank.

Discharge state: MOSFET switches—$M_D$ is turned OFF and $M_S$ is turned ON. Cell1, LC tank, and $M_S$ form a anti-clockwise loop in Figure 4d and LC tank releases the energy stored in Cell1.

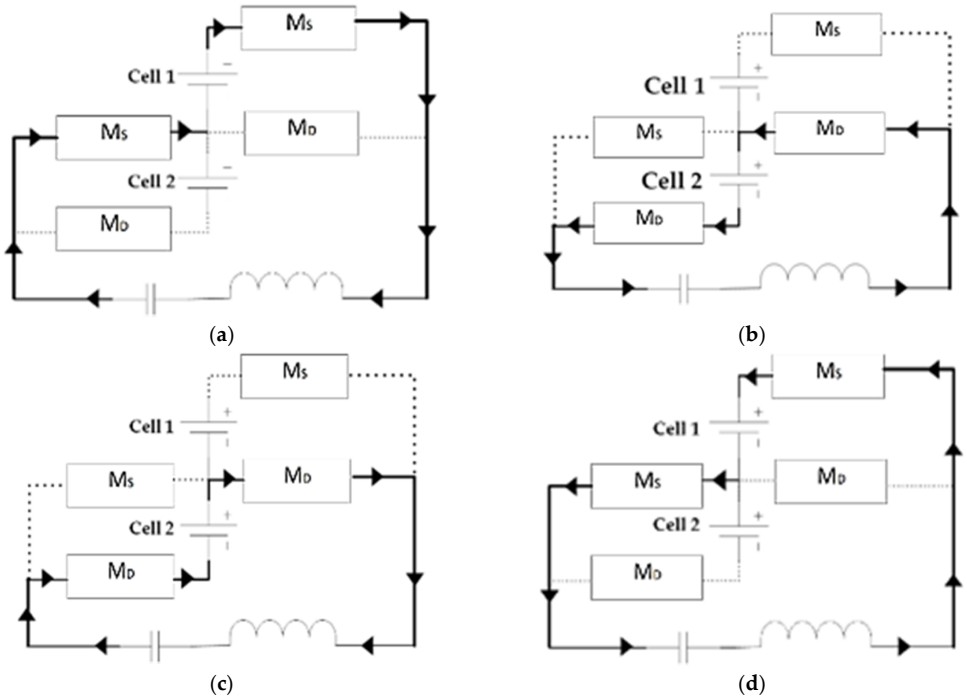

**Figure 4.** Voltage-balancing process in working mode, (**a**) cell 1 store the energy in LC; (**b**) LC release the energy for charge the cell 2; working mode II: (**c**) cell 2 store the energy in LC; (**d**) LC release the energy for charge the cell 1.

## 2.3. Circuit Analysis

Analysis of the Working Mode I flowing rotation is done. LC tank charging state equivalent resistance $R_{eq1}$ can be expressed as:

$$R_{eq1} = 2R_{DS(ON)} + R_{LC} \tag{1}$$

where $R_{DS(ON)}$ is the MOSFET switch, drain-source *ON* state, resistance $R_{LC}$ is the initial resistance of the *LC* tank. When the PWM single is high then apply KVL in this loop

$$R_{eq1}i_1(t) + LC\frac{d^2v(t)}{dt^2} + v_c = V_{cell1} \tag{2}$$

$$i_1(t) = \frac{V_{Cell\,1}}{R_{eq1}} - \frac{V_{Cell\,1}}{R_{eq1}}e^{2\frac{R_{eq1}\,R_c}{LC}(t-t_0)} - \frac{v_c}{R_{eq1}} \tag{3}$$

*LC* tank realize state, equivalent resistance $R_{eq2}$ can be expressed as

$$R_{eq2} = 4R_{DS(ON)} + R_{LC} \tag{4}$$

where $R_{DS(ON)}$ is the MOSFET switch, drain-source *ON* state resistance $R_{LC}$ is the initial resistance of the *LC* tank. When the PWM single is high then apply Kirchhoffs Voltage Law (KVL) in this loop

$$R_{eq2}i_2(t) - LC\frac{d^2v(t)}{dt^2} - v_c = V_{cell2} \tag{5}$$

$$i_2(t) = \frac{V_{Cell\,2}}{R_{eq2}} + \frac{V_{Cell\,2}}{R_{eq2}}e^{-2\frac{R_{eq1}\,R_c}{LC}(t-t_0)} + \frac{v_c}{R_{eq2}} \tag{6}$$

Charge-balancing efficiency of this circuit can be estimated by the equation

$$\eta = \frac{V_{Cell\,2} \times i_2(t)}{V_{Cell\,1} \times i_1(t)} \times 100 \tag{7}$$

In operational condition, consenting to the typical voltage balance of the resonant capacitor $V_c$ accomplished cycle switching,

$$i_L DT - i_L(1-D)T = 0 \tag{8}$$

Here, *D* is the duty cycle, and *T* is the switching time. When the MOSFET switches are turned *ON* and *OFF* by using PWM single then the relation of equivalent circuit current (*i*) with $R_{eq1}$, $R_{eq2}$, and LC tank is shown in Figure 5. According to the Equations (3) and (6) satisfy the equivalent circuit current change and energy transfer configuration.

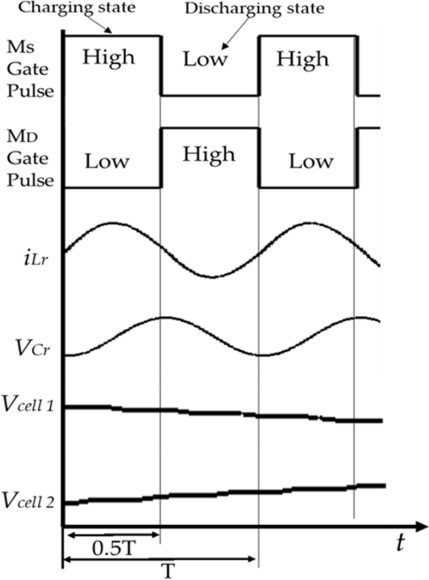

**Figure 5.** Analytical waveforms of MOSFET switches, resonant inductor current capacitor voltage, and cell voltages in the working modes.

## 3. Voltage Balancing Algorithm

The resonant converter method is an active balancing method. The voltage sensors are used as monitoring intergrade circuits (IC) to measure the battery cells' status. All sensor values are read by the monitoring IC and are communicated with the microcontroller. The microcontroller assembles the IC's data, then estimates the voltage of individual battery cells and compares the value, detects the higher voltage cell and lower voltage cell, and sends the instruction to activate the associated switch for making an appropriate path to connect with LC resonant converter. The DC-DC converter carries out voltage balancing as per the definition of the proposed algorithm. The proposed voltage-balancing algorithm flowchart is shown in Figure 6. Based on the battery cell voltage IC reading, the algorithm can decide for operating the balancing converter. In battery string, the unbalance cells charge, overcharge, and undercharge in battery cells charge are included in this voltage-balancing algorithm. The proposed charge-balancing is presented in Algorithm 1.

---

**Algorithm 1** Proposed Charge Balancing

---

1: Initialize the system;
2: Read the battery cells voltage status;
3: Check the cells' status, so whether battery cells are normal or abnormal.
4: **If** (battery cells are operating at normal),
5:   go to step2
6: **else** (Battery cell abnormal)
7:   Go to next
8: Check the condition of the battery cell;
9: **If** (cell in the battery is found as overcharged)
10:   Execute the condition of the overcharge limit of the battery cell;
11:   Go to step 7;
12: **else** (a cell in the battery is found as undercharged)
13: Execute the condition of the undercharge limit of the battery cell and
14: Start the cell balancing process.
15: Estimate the voltage of the equivalent cells.
16: Check the voltage value by IC for discharge balancing.
17: Execute the cell discharge mode if; the battery cell voltage is classified as overcharged,
18:   where voltage is the highest IC reading value of the battery cells.
19: Control the MOSFET switches by PWM signal to detect battery cells for discharging the overcharged, control the DC-DC resonant converter, and allow the balancing current for discharging.
20: Check the balancing of discharging cells.
21: **If** (equivalent cell is unbalanced && voltage is lower IC reading value of all battery cells)
22:   Go to step7
23: **else** (the cells in the battery balanced)
24:   Go to step 2
25: Check the cell voltage value for charge balancing.
26: **If** (battery cell voltage is classified as an undercharged battery && Voltage is the IC reading state)
27:   Execute the charge mode.
28: **else** Go to step 2.
29: Control the MOSFET switches by PWM signal to detect battery cell for charging the undercharged, control the DC-DC resonant converter, and allow the balancing current for charging.
30: Check the balancing status of the charging cell.
31: **If** (corresponding cell is unbalanced && Vvoltage is higher IC reading value)
32:   Go to step 7;
33: **else** (cells in the battery is balanced)
34:   Go to step 2;
35: Execute the voltage-balancing process
36: Repeat

---

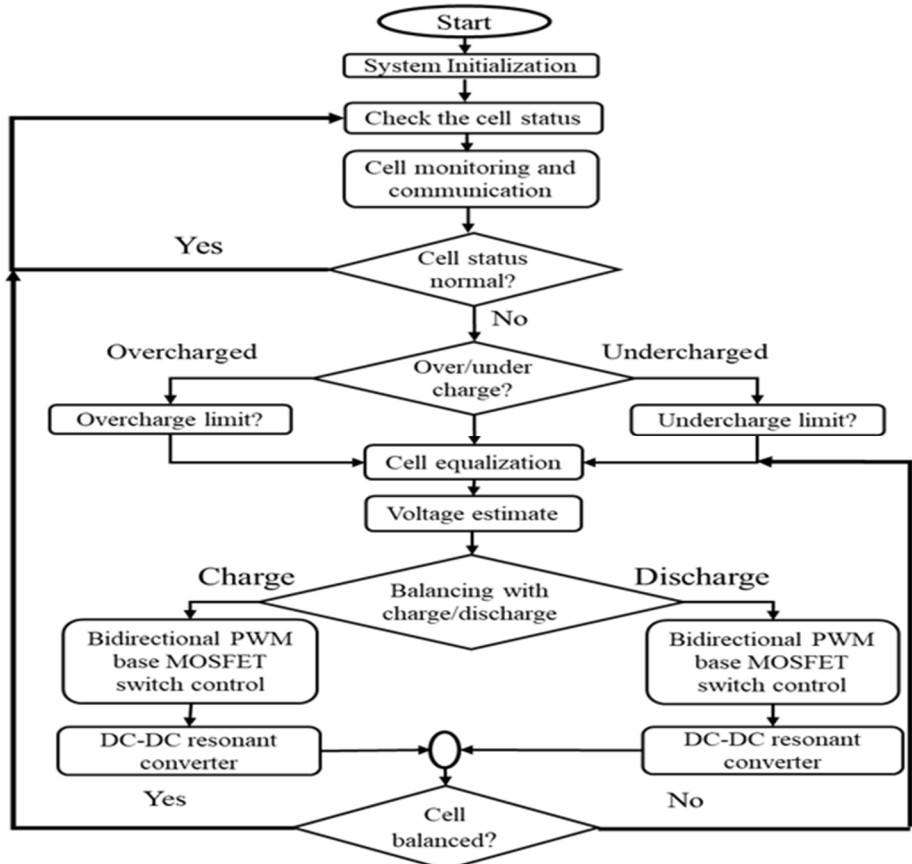

**Figure 6.** Flowchart of the proposed charge equalization algorithm.

## 4. Simulation

Two Li-ion battery cell base active voltage-balancing circuits are simulated in MATLAB SIMULINK-2016a software. To execute the simulation, we used the ideal component from the Simulink library. In the simulation, two 100 F capacitors are considered, two Li-ion battery cells, and the initial voltage of the two capacitors are 2.7 V and 2.5 V, respectively. The resonant tank switch capacitor is 222 μF with an initial voltage of 0 V, and the inductor is 100 μH. A pair of pulse-generator complementary signals with a 50% duty cycle controls all MOSFET switches, and this working principle is proposed in Section 2.

Figure 7a shows the MOSFET switches, the gate pulse signal, resonant inductor current, and capacitor voltage. The switching frequency (*fs*) is 1068 Hz, equal to the resonant frequency (*fr*) for that circuit to achieve soft switching. In the circuit, the charging and discharging state's current amplitude is the same oscillation as in the resonant tank. There is no switching voltage drop and circuit loss. For this, voltage-balancing time depends on the resonant inductor and capacitor. Figure 7b shows the MOSFET switch, gate pulse signal, resonant inductor current, and capacitor voltage. The switching frequency is half of the resonant frequency that is 534 Hz. In the circuit, the resonant tank oscillation's current amplitude is not the same in the charging state and discharging state. The current is interrupted during the operation time circuit and leads to the damped resonance oscillation and circuit loss.

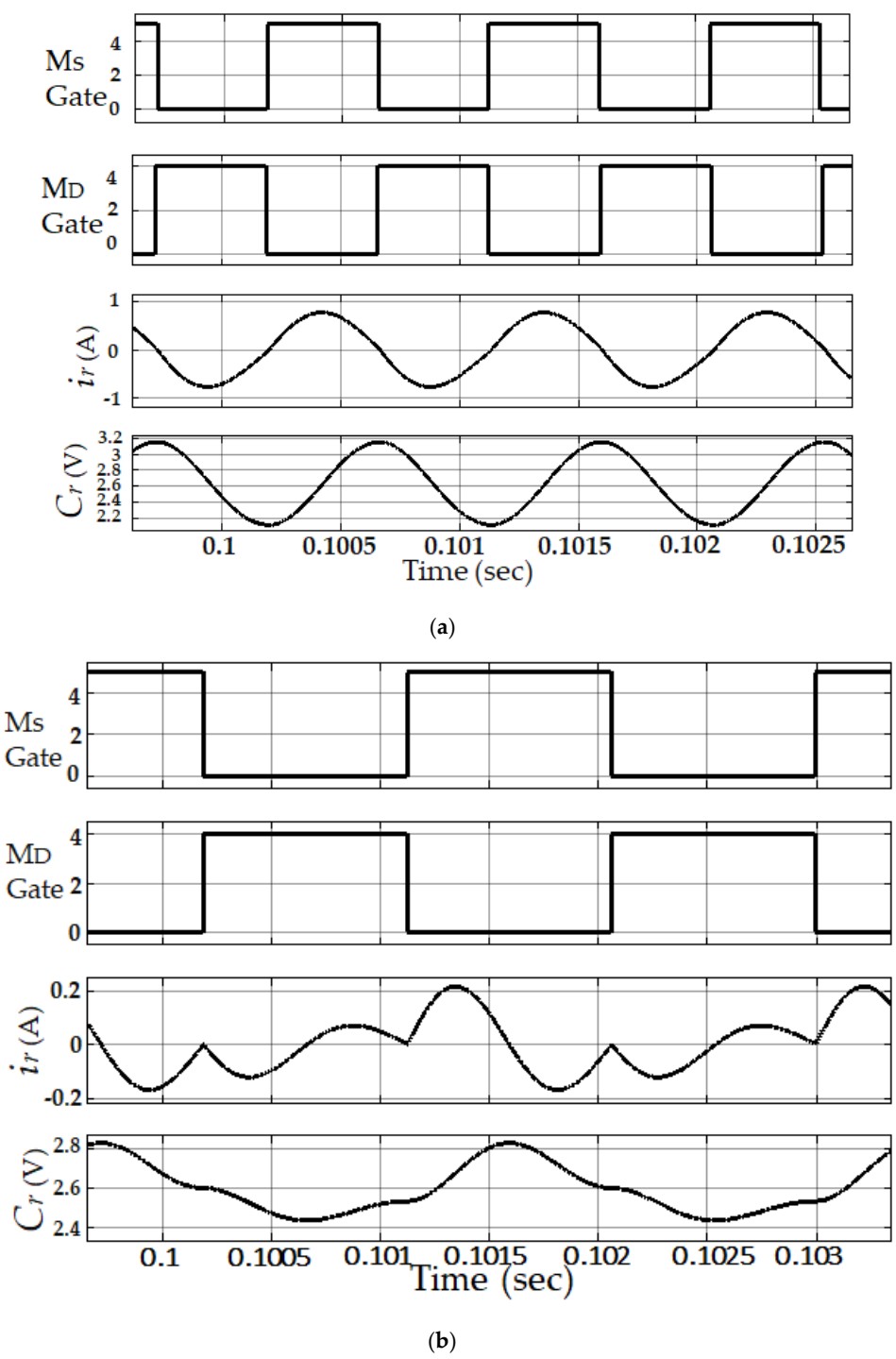

(a)

(b)

**Figure 7.** *Cont.*

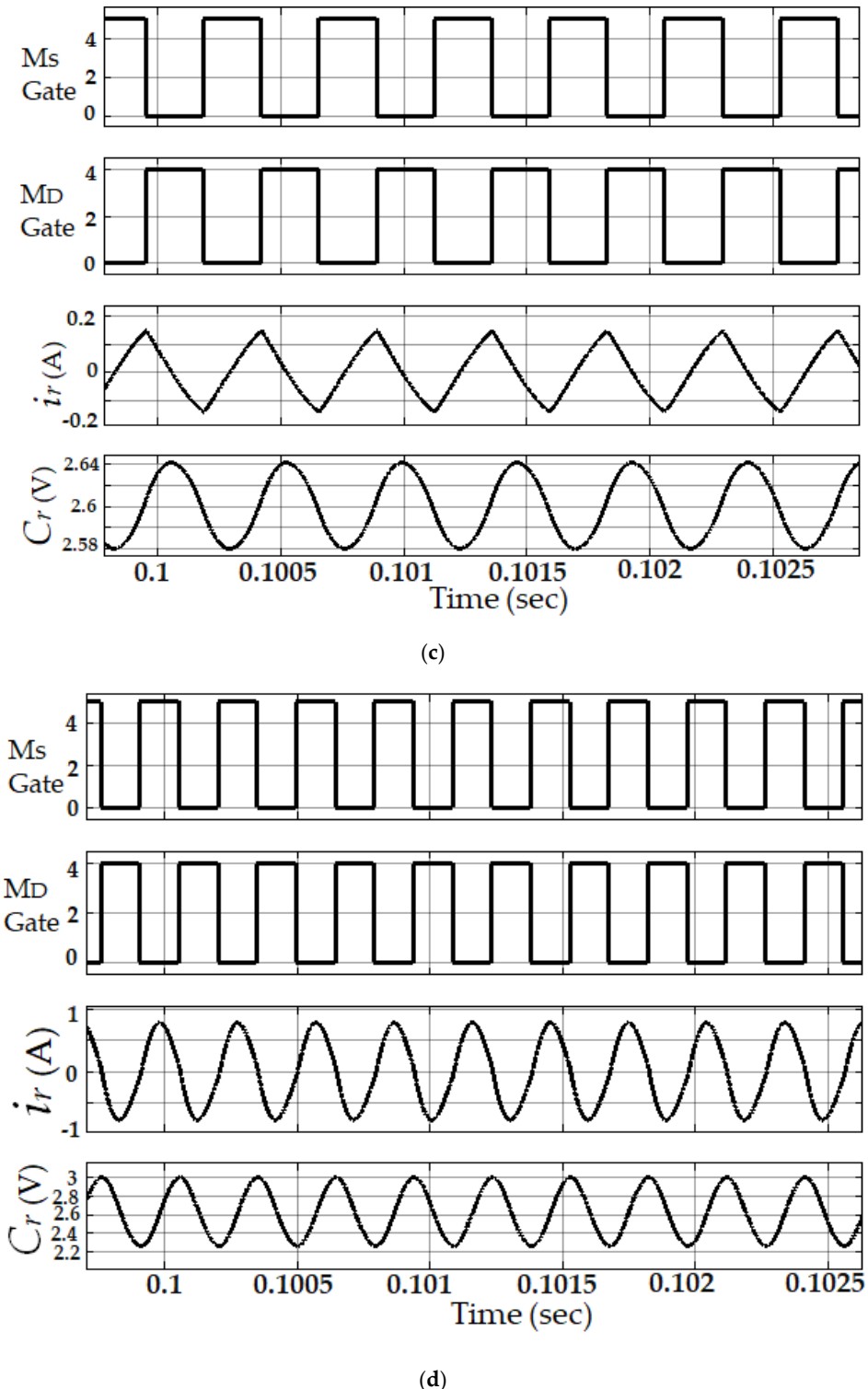

(c)

(d)

**Figure 7.** Simulation waveforms with an internal resistor, (**a**) *fs* = *fr*, (**b**) *fs* = 0.5 *fr*, (**c**) *fs* = 2 *fr*. (**d**) Smaller inductor and capacitor.

When the switching frequency (2136 Hz) is double resonant, then MOSFET switches have not achieved the soft switching as shown in Figure 7c. This is because resonant inductor magnetizing current is not enough, and additionally, voltage stress occurs in the switches. There are switching voltage drop, and circuit loss for this circuit takes a long balancing time. Figure 7d shows the MOSFET switches, the gate pulse signal, resonant inductor current, and capacitor voltage. In this simulation,

the LC tank switches capacitor and inductor are changed to 100 µF and 22 µH. Figure 7a shows that the resonant current wave is larger than that in Figure 7d, but the charging and discharging state's current amplitude is the same oscillation as in the resonant tank. For this enormous current wave, massive energy transfer takes place. In the LC tank, the inductor and capacitor have small capacitance, hence the resonant frequency increases and a comparably small amount of energy storage in the LC tank increases the balancing time between the two cells. Though there is no switching voltage drop, the circuit loss increases. This circuit takes a long balancing time.

Figure 8 shows the simulation voltage-balancing result. Initially, the voltage difference between the two super-capacitors was 200 mV. The voltage balancing operation principle was the same as in Figure 7. After the balancing operation, the voltage difference decreased. At the same time, the resonant inductor current and switch capacitor's voltage oscillation amplitude reduced. After about 143 seconds, the voltage difference becomes 0 V, and the resonance stops.

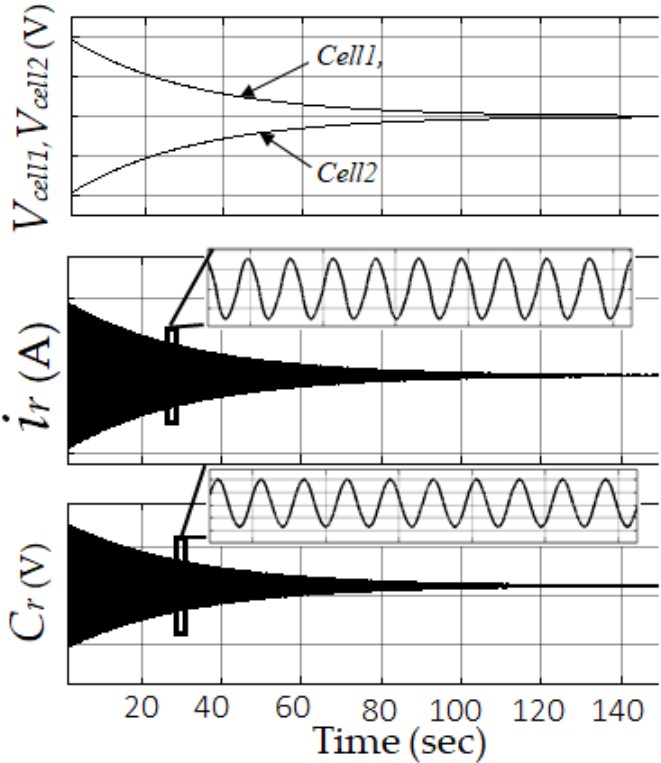

**Figure 8.** Simulation result of the voltage balancing system.

## 5. Experiment Result and Discussion

### 5.1. Implementation

An experimental approach is conducted to verify the theoretical and simulation results. Figure 9 shows the schematic diagram of the proposed voltage-balancing system and the experimental setup for battery cells built in the laboratory using the parameters given in Table 1. Two and four battery cell-balancing circuit prototypes are built in the laboratory. In this circuit, Ms is a single MOSFET (IRF540A) switch, and $M_D$ is a bidirectional MOSFET (IRF8721) switch. All MOSFET switches operate on a 50% duty cycle of the PWM signal from the microcontroller.

### 5.2. Experiment Result

Figure 10 shows the resonant tank waveform in different switching conditions. When the resonant inductor is 100 µH, and the resonant capacitor is about 222 µF, the switching frequency is 1068 Hz, close to the resonant frequency, as shown in Figure 10a.

**Table 1.** List of the component.

| Part | Part Name | Value |
|---|---|---|
| Single switches ($M_S$) | nMOSFET | IRF450A |
| Bidirectional Switches ($M_D$) | nMOSFET | IRF8721 |
| Gate driver | Optocoupler | 817c |
| | Logic gate | SN7404 |
| Resonant tank | Inductor | 100 μH |
| | capacitor | 222 μH |
| Microcontroller | Arduino Uno | atmega328p |
| Monitoring IC | - | Different amplifier |
| Battery | Li-ion | 4200 mAh, 3.7 V (Ultra Fire BRC 18650) |
| | - | |
| | Lade-acid | 12 V, 1.2 Ah (Battery mart) |

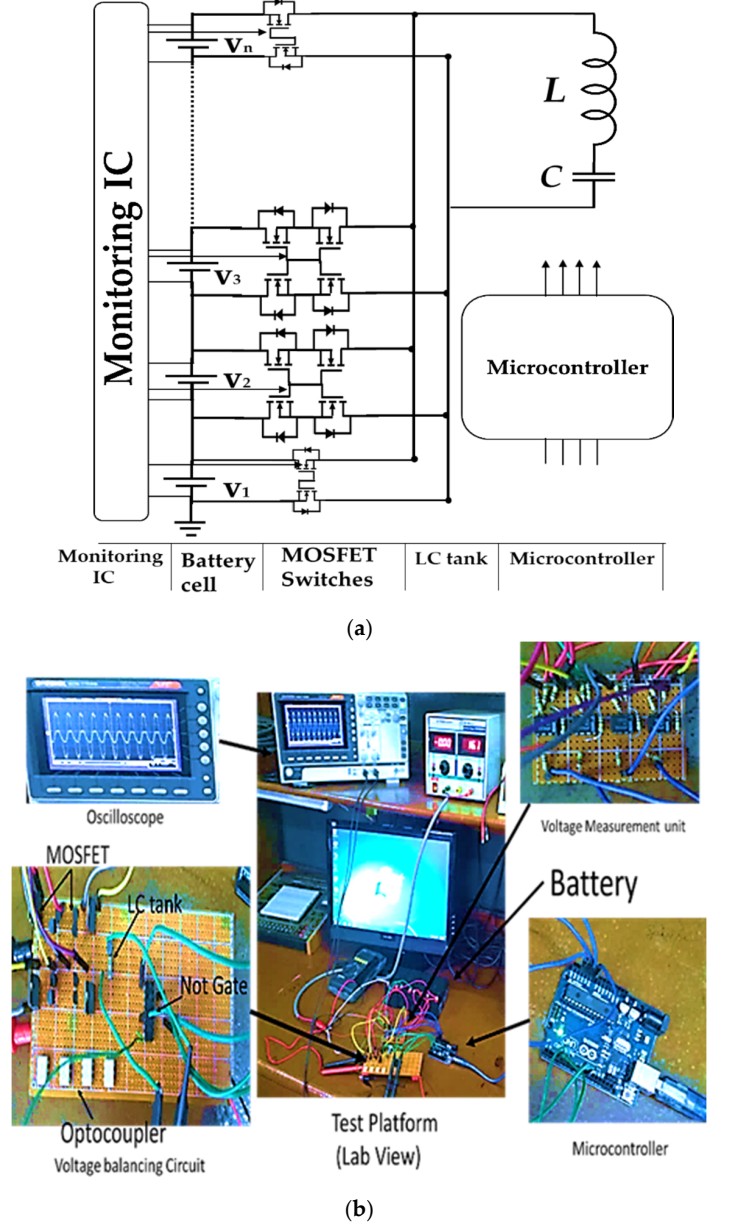

(a)

(b)

**Figure 9.** Hardware development: (**a**) schematic diagram of the balancing circuit and (**b**) experimental setup in a laboratory.

Figure 10a shows that the resonant capacitor voltage and current amplitude and wavelength are the same in each charging and discharging state with switching amplitude and wavelength, respectively. It meets the zero-voltage gap conditions. Figure 10b shows the enactment when the switching frequency value increases to double the resonant frequency than the waveforms. Suppose the resonant capacitor voltage and current wavelengths increase, and amplitude changes in each charge and discharge state with swathing amplitude and wavelength. The simulation analysis is consistent with experimental analysis. When the resonant inductor is 22 µH, and the resonant capacitor is about 100 µF, the switching frequency is 3.39 kHz, close to the resonant frequency, as shown in Figure 10c. The resonant capacitor voltage and current wavelength are smaller than Figure 10a. When the switching frequency is set equal to the resonant frequency, then the resonant tank's current will be quasi-sinusoidal, as shown in Figure 10a,c. If the resonant frequency is small, then the balancing time is shorter because the resonant capacitor voltage wavelength became larger.

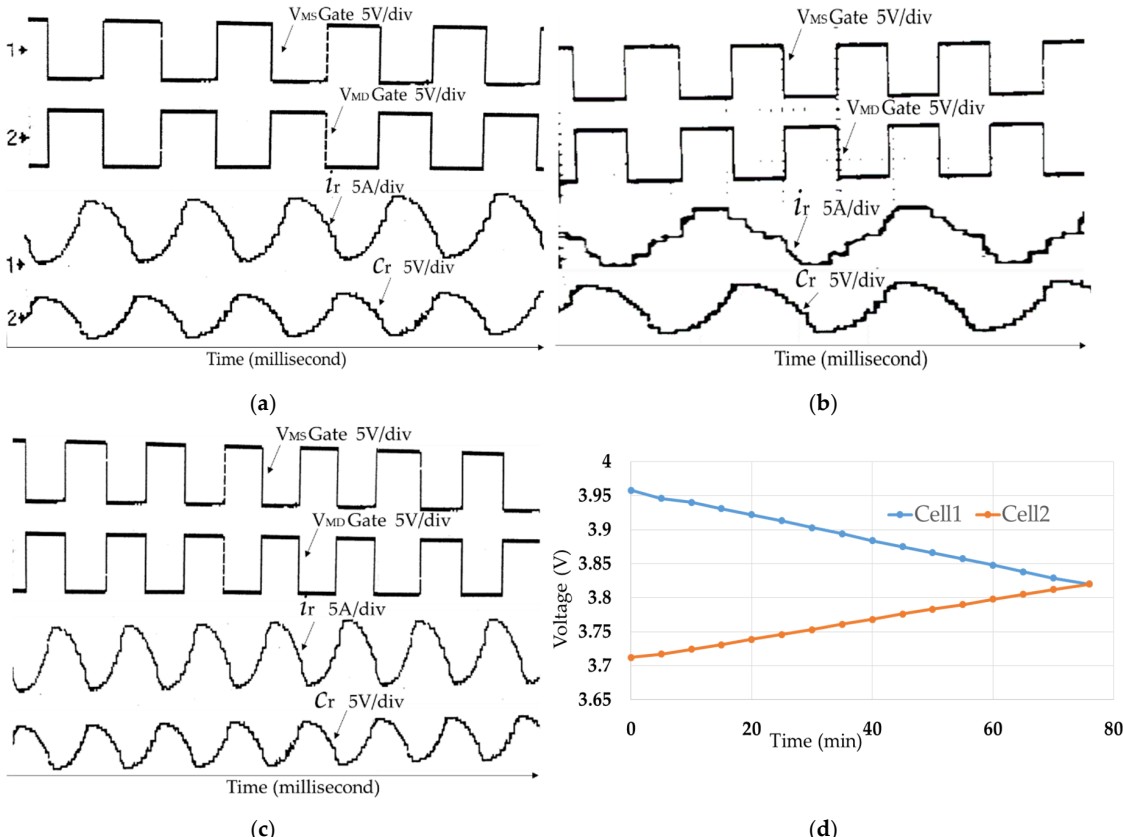

**Figure 10.** Experimental result waveforms with an internal resistor, (**a**) $f_S = fr$, (**b**) $f_S = 2\,fr$, (**c**) small inductor and capacitor, (**d**) measured balancing result for two Li-ion battery cell.

First, two Li-ion battery voltage-balancing processes are shown in Figure 10d, and the rated voltages of the cells' are 3.7 V. The initial voltages of the battery cells are 3.958 V and 3.712 V, respectively. After 76 minutes, with 94.8% efficiency (using equation 7), the voltage difference of the battery cells becomes zero, and the balancing time depends on the cells' capacity. If the cell's capacity is more extensive then the balancing time is longer. Three Li-ion battery cells are the base voltage-balancing processes as shown in Figure 11a. The battery cells' initial voltages are 3.98 V, 3.89 V, and 3.66 V, respectively. After 90 minutes, the voltage difference between the battery cells becomes zero. Second, to check the performance for a different kind of electrochemical battery cell, a lead-acid (12 V, 1.2 Ah) based experiment is conducted for four cells in the laboratory. Initially, the voltage difference was 400 mV, and the balancing circuit takes 199 minutes to balance the voltage difference and achieve 0-V gaps as shown in Figure 11b. Figure 12 shows a cyclic test performed under the charge/discharge

condition at current 1 A. For two battery cells OCV was 3.92 V and 3.843 V, respectively, and balanced after 80 minutes. Both cells are charge/discharge after 210 min, and balanced cells are elongated in the charge time.

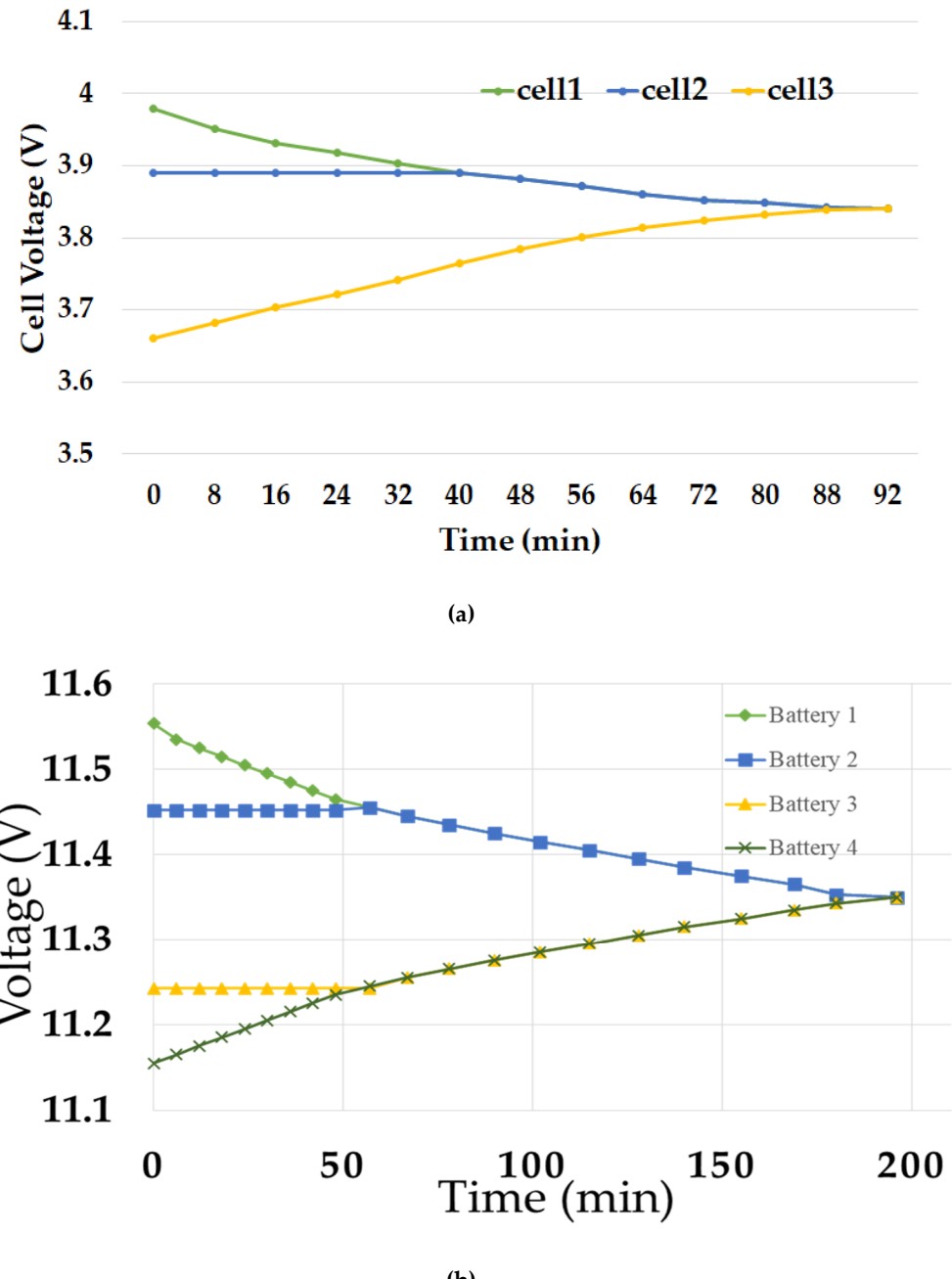

(a)

(b)

**Figure 11.** The experimental result, (**a**) measured for three Li-ion battery cells, (**b**) measured for four lead-acid battery cells.

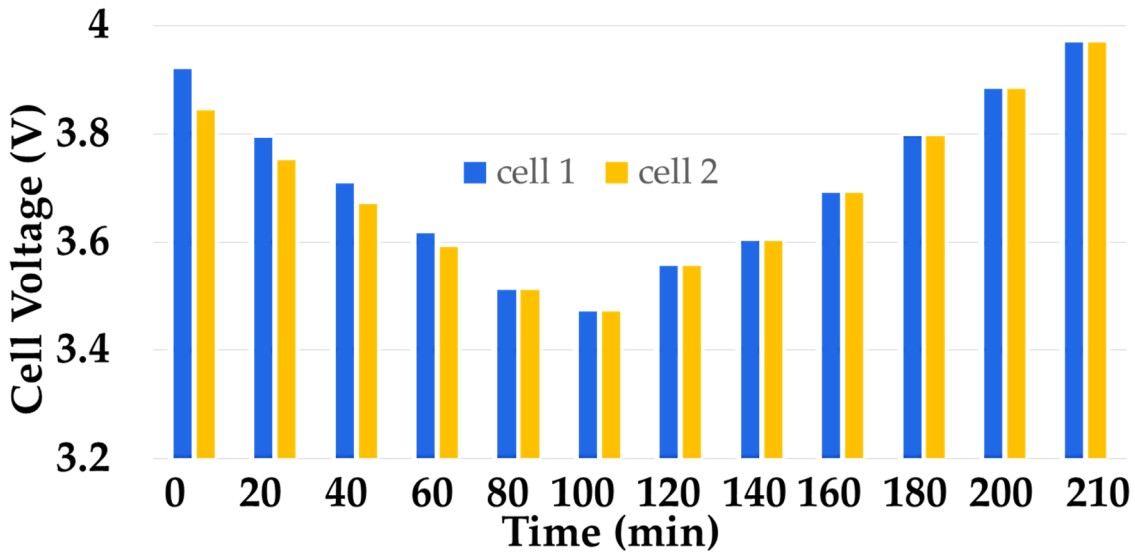

**Figure 12.** Experimental result: measured for two Li-ion cells balancing circuit test under discharging/ charging time. ($I_{dis} = I_{ch} = 1$ A).

*5.3. Benchmark*

The proposed balancing circuit and different existing balancing circuit result comparisons are shown in Figure 13. For this comparison, we choose some research work to benchmark with our proposed circuit from 2014 to 2019. This chosen research work was based on the single transformer, quasi-resonant, and single series resonant circuits similar to our research work. Also, to select the two battery cell-based balancing circuit results to improve the proposed circuit can be easily illustrated. The proposed circuit takes less time to balance the cell voltage difference, recover the balancing energy, and archive the 0-V gap.

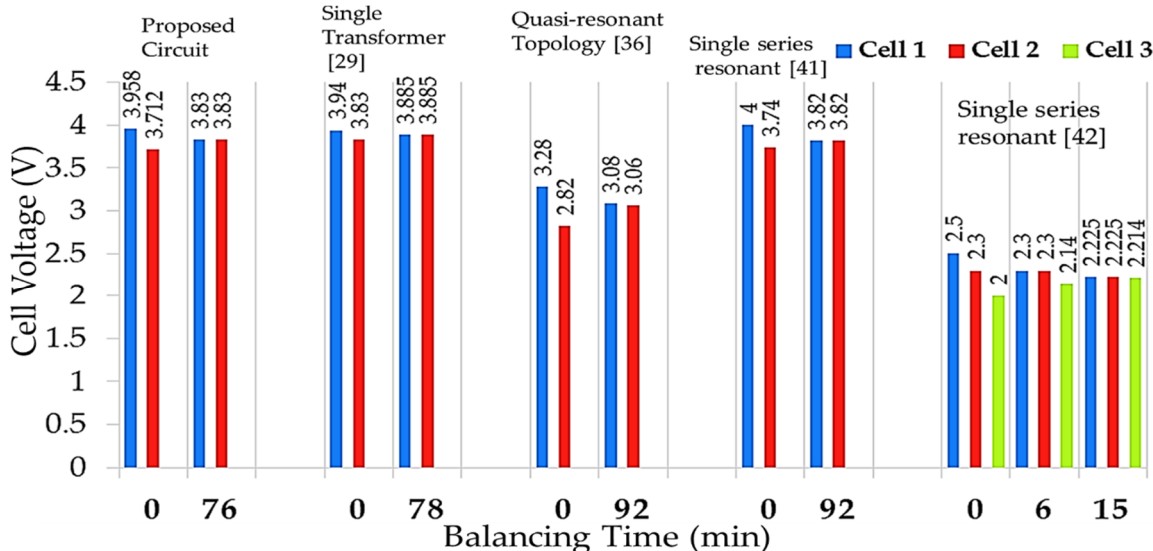

**Figure 13.** Balancing circuit result comparison: proposed circuit to other works.

Lee et al. [29] used a single transformer to transfer energy. In this circuit, a 4400-mAh Li-ion battery is used. Initially, cell voltage was 3.94 V and 3.83 V. The voltage difference between the two cells was 110 mV, and theoretically, cell voltage balance was 3.885 V but was due to magnetizing loss in the transformer. The cell voltage is balanced in 3.87 V and takes 78 minutes. Shang et al. [36] used quasi-resonant and boost topology to transfer the energy. In this circuit, two LiFePO4, 6.2 Ah batteries

are used. Initially, cell voltages were 3.27 V and 2.825 V. The voltage difference between the two cells was 445 mV, and theoretically, the cell voltage balance was 3.05 V, but after 92 minutes, the cell did not balance, and it had 20-mV voltage gaps.

Lee et al. [41] proposed a single LC energy carrier to transfer energy. In this circuit, a 4400-mAh Li-ion battery is used. Initially, cell voltages were 4 V and 3.74 V. The voltage difference between the two cells was 260 mV, and theoretically, cell voltage balance was 3.87 V, but due to power loss in MOSFET resistance, the cell voltage balance was 3.82 V and takes 92 minutes. The control system of this circuit is much complicated. Yu et al. [42] also used a single LC energy carrier to transfer energy. In their study, they used three 300 F super-capacitor (SC). Initially, SC voltage was 2.5 V, 2.3 V, and 2.0 V. The voltage difference between three SCs was 500 mV, and theoretically, cell voltage balance is 2.27 V, however due to 1% death time in switching and internal MOSFET resistance, the cell voltage balance is 2.26 V with 10-mV voltage gaps after 16 minutes.

In this proposed circuit, a single LC energy carrier is used to transfer the energy. In our circuit, we used very low resistance bi-directional MOSFET switches and a large capacitor and inductor to store the energy. To drive the MOSFET switches, we used the PWM signal and the switching frequency remains the same as the resonant frequency. For the experiment, a 4200-mAh Li-ion battery is used. Initially, cell voltages were 3.958 V and 3.712 V. The voltage difference between the two cells was 273 mV, and theoretically cell voltage balance is 3.85 V, and it is balanced in 3.83 V and achieved zero voltage gap.

### 5.4. Discussion

The proposed voltage-balancing circuit is compared with the standard converter-based balancing circuit on the several key points illustrated in Tables 2 and 3. The comparison study is described in Table 2 based on the balancing circuit, voltage balancing time, execution difficulty, energy flow, control complicity, power loss, voltage and current stress, efficiency, cost, and size for n-cell battery voltage balancing circuits. The quantitative balancing performance comparison between conventional and proposed circuits is depicted in Table 3.

**Table 2.** Comparison along proposed voltage balancing circuit.

| Type<br>Parameter | Boost<br>Converter<br>[15,36] | Buck-Boost<br>Converter<br>[37] | Fly-Back<br>Converter<br>[46] | Ramp<br>Converter<br>[15,40] | Cuk<br>Converter<br>[34] | Resonant<br>Converter<br>[41,42] | Proposed<br>Converter |
|---|---|---|---|---|---|---|---|
| Switch's | n + 1 | 2n | 2n + 6 | n | n | 2n | 2n − 2 |
| Diode | 0 | 0 | 2 | n | 0 | 1 | 0 |
| Inductor | 2n − 2 | N − 1 | 0 | N − 2 | n | 1 | 1 |
| Capacitor | N − 1 | 0 | 2 | n | N − 1 | 1 | 1 |
| Transformer | 0 | 0 | 2 | 0 | 0 | 0 | 0 |
| Power loss * | 1.9% $^{\upsilon}$ | 22% $^{\upsilon}$ | 7.2% | 9% | 6.2% | 13.6% [41]<br>3.6% [42] | 4.9% |
| Efficiency | 98% | 82.5% | 92% | 87.4% [40] | - | 78.9% [41]<br>96% [42] $^{\upsilon}$ | 94.8% |

\* Based on voltage balancing experimental result between the cells, $^{\upsilon}$ voltage gap between the cell after equalization.

Inside the Cuk converter, cell voltage-balancing speed and efficiency is high, with low current and voltage stress; however, many switches are used. It needs a complex control system and requires accurate voltage sensing [31,32]. The balancing efficiency and speed are high with minor power loss, lower current, and voltage stress in the boost converter-balanced circuit; nevertheless, it requires appropriate voltage sensing, an intelligent and complex control system, and is costly [33,34].

**Table 3.** Balancing performance comparison between conventional and the proposed circuit.

| Type Parameter | Buck Boost Converter [39] | Fly-Back Converter [44] | Cuk Converter [33] | Quasi-Resonant Converter [36] | LC Matrix Converter [21] | Resonant Converter [42] | Proposed Converter |
|---|---|---|---|---|---|---|---|
| Battery/SC Rating | 4.0 Ah Batteries | 2.65 Ah Batteries | 10.0 Ah Batteries | 6.2 Ah Batteries | 6.2 Ah Batteries | 300 F SC | 4.2 Ah Batteries |
| Initial voltage Difference (mV) | 560 | 250 | 1000 | 620 | 500 | 527 | 246 |
| Final voltage gap (mV) | 0 | 25 | 0 | 80 | 4.5 | 10 | 0 |
| Balancing time (s) | 2700 | 5500 | 4500 | 6800 | 3500 | 900 | 4680 |
| Normalized voltage balancing time (s/j) | 20.1 | 15.4 | 7.5 | 20.3 | 12.3 | 11.2 | 13.8 |

J = Joule, s = second, mV = millivolt, SC = super capacitor.

On the contrary, in the buck-boost converter-based balancing circuit the balancing speed is very high, and shows better efficiency with lower voltage and current stress; nevertheless, it requires complex control systems, many switches, and is costly [32,35–37]. In ramp converter, battery cell balancing speed is medium and shows good efficiency with less power loss; however, it is medium current and voltage stress, costly, and extensive [38]. The balancing speed and efficiency are good with the low power loss, low current and voltage stress in the resonant converter circuit; however, it needs a complex control system and is medium in size and costly [39,40]. On the contrary, the balancing speed is medium with high efficiency, low current, and voltage stress; nevertheless, it is extensive and costly [41,42]. The proposed cell balancing has a high balancing speed and efficiency with low power loss. The voltage and current stresses are low, small in size because we reduced the number of switches, diodes, and the associated component of LC energy carrier compared with [41,42]. It needs less balancing components compared to other equalizers. However, it needs intelligent control. The proposed cell-balancing circuit is a bidirectional voltage-balancing circuit for the battery cell charge and discharge balancing and could be applicable for battery storage balance in EV with a modular design.

## 6. Conclusions

A battery voltage-balancing algorithm and circuit are presented in this paper. This system can be performed in Li-ion battery cells, supercapacitors, and other rechargeable batteries for EV applications. The development of the operating principle, the algorithm, and prototype performance for the proposed voltage-balancing system is provided. Thus, circuit loss will decrease, and the balancing circuit takes a short balancing time (for two 4200 mAh, 3.7 V Li-ion cells, it takes 76 minutes). This voltage-balancing control system has exciting properties. It has successfully reduced the circuit loss, increasing the battery voltage balancing efficiency, and extending the battery cells string capability and life. The simulation and experimental approach is conducted to substantiate the theoretical analysis for the proposed balancing circuit. Battery voltage-balancing time is dependent on the design of the resonant tank. The battery cell's energy storage capacity is relational and inversely proportional to the switched capacitor and switching frequency. In this study, the temperature variation was not considered. In the future study, temperature variation will be considered and presented.

**Author Contributions:** Conceptualization, A.A.H. and M.K.H.; methodology, A.A.H.; software, A.A.H. and S.I.; validation, A.A.H., M.K.H., and E.H.; writing—original draft preparation, A.A.H.; writing—review and editing, M.K.H., A.T.A.G., S.I., and E.H.; visualization, S.I. and E.H.; supervision, M.K.H.; project administration,

M.K.H.; funding acquisition, M.KH. and A.T.A.G. All authors have read and agreed to the published version of the manuscript.

**Funding:** This work was supported in part by the National University of Malaysia under Grant GGPM2020-028 and GGPM-2019-063.

**Conflicts of Interest:** The authors declare no conflict of interest.

## Abbreviations

**Nomenclature**

| | |
|---|---|
| $\eta$ | Efficiency |
| BMS | Battery management system |
| $C$ | Capacitor [V] |
| C2C | Cell-to-cell |
| C2P | Cell-to-pack |
| CO | Carbon-mono-oxide |
| $CO_2$ | Carbon-di-oxide |
| $D$ | Duty cycle |
| DC | Direct current |
| EV | Electric vehicle |
| $i$ | Current [A] |
| IC | Intergrade circuit |
| ICE | Internal combustion engine |
| $i_L$ | Inductor current [A] |
| $L$ | Inductor [H] |
| LC | Resonant tank |
| MD | Bi-directional MOSFET |
| MS | Single MOSFET |
| MOSFET | Metal Oxide Silicon Field Effect Transistor |
| OCV | Open circuit voltage |
| P2C | Pack-to-cell |
| P2P | Pack-to-pack |
| PWM | Pulse-width modulation |
| $R_{ds}$ | Internal drain to source resistance in the MOSFET [$\Omega$] |
| $R_{eq1}$ | LC tank charging state equivalent resistance [$\Omega$] |
| $R_{eq2}$ | LC tank discharging state equivalent resistance [$\Omega$] |
| SOC | State of charge |
| $t$ | Time [s] |
| $t_0$ | Initial time [s] |
| $T$ | Switching time [s] |
| $V_c$ | Resonant capacitor voltage [V] |
| $V_{Cell}$ | Battery cell voltage [V] |

## Appendix A

Single capacitor, inductor, and transformer-based balancing circuits balance the cell voltage between two cells in a string. Here, 2n bidirectional switches or 2n + 2n diodes are used with a single capacitor, inductor, or transformer. These balancing circuits are faster and more efficient than passive balancing. However, they require a complex control system, ripple current, and magnetizing loss. Multi capacitor, inductor, and transformer-based balancing circuits balance the cell voltage between two adjoined cells in the string. Here, 2n single switches are used for n cells. These balancing circuits have good balancing speed, medium complexity. However, they have high ripple current, magnetizing, and core loss.

Cuk converter consists of two switches, a capacitor, and two inductors that are applicable for two cells. The charge-balancing speed is high but the control system is complicated and faces magnetizing loss. In a boost, the converter inductor is used in parallel that can transfer the energy from any cell to

any cell in the string. It can transfer the excesive energy from higher to lower capacitive cell between adjoining cells or any cell to cell, but this circuit faces magnetizing loss and ripple current. Buck-boost converter is the combination of buck and boost converter and works like a multi inductor-based balancing circuit. This circuit has good energy efficiency but requires smart voltage sensing and a complex control system.

Ramp converter is an upgrade version of the multi-winding transformer-based balancing system. In this converter, the transformer secondary winding is connected with the cells, and the primary winding is connected with a single-cell through of voltage-detection circuit. This balancing circuit has a complex control system and hardware implication. The resonant converter is one of the most suitable charge balancing circuits. This circuit can transfer the energy from higher cell to lower cell between adjoining cells or any cell to any cell. This balancing circuit has a high balancing speed, efficiency, and low power loss. However, it requires a complex control system. The flyback converter is also used in the cell-balancing system in high power applications. It is also used in P2P, P2C, C2P balancing system. It has a high balancing speed and is efficient but requires smart voltage sensing, intelligent control, and faces core magnetizing loss.

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
