# Peer review of "Resonant Energy Carrier Base Active Charge-Balancing Algorithm"

_electronics, doi:10.3390/electronics9122166_

Round 1

Reviewer 1 Report

this paper is interesting for but need some major revision 

1- English need to improve 

2- quality pf pictures are not clear. change all of them.

3- paper title need to change based on the paper. For example why you didn't mention Electric Vehicle in your title?

Author Response

Reviewer 1: This paper is interesting but needs some major revision.

Concern 1: English needs to improve.

  Author Response: We have resolved the grammatical issues from the manuscript. English native speaker has revised the paper

 Concern 2: The quality of the pictures is not clear. Change all of them.

Author Response: We have improved the picture quality based on your suggestion.

Concern 3:

 The paper title needs to change based on the paper. For example why you didn't mention Electric Vehicle in your title?

Author Response: In this paper, we proposed Resonant Energy Carrier-based balancing circuit for Battery Management System (BMS) that applies to low power applications such as laptops, UPS, IPS, Home Solar energy storage systems as well as high power application such as electric vehicles, micro-grid. Therefore, we chose this title for our manuscript.

Reviewer 2 Report

Look at my all comments provided in the attachment and address them carefully.

Author Response

Reviewer 2: This paper presents “Resonant Energy Carrier Base Active Charge Balancing Algorithm”. The manuscript is well organized and written. Nomenclature, subscript, superscript, acronyms, and abbreviations are big issues in this manuscript. According to my opinion, this paper is accepted with minor comments. In addition, the authors are suggested to address the following comments in order to meet the requirements of the Journal.

Concern 1: Define Nomenclature, Greek symbols, subscripts, superscripts, and acronyms separately in table form.

Author Response: We have added the table based on your suggestion.

 Concern 2: . Define abbreviations such as MOSFET, HEV, C2C, P2C, C2P, OCV, SOC, EV, ECM, BMS, BTMS, and so on separately in table form.

Author Response:  We have added the table base on your suggestion.

Concern 3: Define C-rate first time when you use in the manuscript. We all know that C-rate is the measurement of the charge and discharge current with respect to its nominal capacity. Define in this manuscript so that readers can understand.

 Author Response: Based on the reviewer's suggestion we have included the cell details in table 1 so that reader can easily access the battery cell configuration.

Concern 4: Check spacing between value and deg C symbol throughout the manuscript.

Author Response: We have corrected as per the comment.

 Concern 5: In Literature review, only two or three papers are discussed about algorithm. Add below relevant references in the Introduction section about NN.

· VG Choudhari, AS Dhoble, S Panchal, “Numerical analysis of different fin structures in phase change material module for battery thermal management system and its optimization”, International Journal of Heat and Mass Transfer 163, 120434, 2020.

· MK Tran, A Mevawala, S Panchal, K Raahemifar, M Fowler, R Fraser, “Effect of integrating the hysteresis component to the equivalent circuit model of Lithium-ion battery for dynamic and nondynamic applications, Journal of Energy Storage 32, 101785, 2020.

 Author Response: We have included the suggested relevant references in the manuscript.

 Concern 6: Check Eq 3 and Eq 6.

Author Response: We have revised Eq. 3 and Eq. 6.

 Concern 7: Provide the actual picture of experimental set-up.

 Author Response: We have provided an actual picture of the experimental set-up that was built in laboratory that tested at the experimentation.

 Concern 8: What was the charging, constant-current and constant voltage (CC-CV) or constant current (CC)?

 Author Response: The charging and discharging current was 1A that already mentioned in 293 and 294 lines for this experiment.

 Concern 9: Did authors use any thermocouple during experiment? Please provide the temperature plot.

Author Response: During the experimental period in the laboratory it was normal room temperature (24-25° C that was covered air condition) so that we did not use thermocouple for MOSFET switches and we used normal connecting wire. 

 Concern 10: Which type of thermocouple did the authors use for surface temperature measurement? T-type or K-type? Provide details with an accuracy.

 Author Response: we did not use thermocouple for MOSFET switches and we have used normal connecting wire. 

 Concern 11: Provide the technical specification of battery cell used for the experimental work, such as nominal voltage, nominal capacity, anode, cathode and electrolyte material.

 Author Response: We give the battery cell technical specification in table 1.

 Concern 12: From Fig 3 onwards, font size inside all figures are too small. Increase it.

 Author Response: We have corrected by improving the text size and resolution for figures.

 Concern 13: In this manuscript, there is no experimental uncertainty. It is always recommended to present “Experimental Uncertainty” analysis if you work only with experiments.

 Author Response: In the battery management system, cell balancing circuits are used to protect battery cell from the hazard, also the previous researchers have not considered the experimental uncertainty for cell balancing circuit. Based on your recommendation we have pointed the “Experimental Uncertainty” analysis for future study in the conclusion section.    

Concern 14: In conclusion, give only main findings of your research with an appropriate value.

Author Response: Our proposed circuit can work of different energy storage system like as Li-ion battery, Lead acid battery so that we discussed.

Author Response:  Based on your suggestion we highlight our main finding in conclusion that is “for two 4200 mAh, 3.7 V Li-ion cells it takes 76 minutes”.

 Concern 15: Check ref [5].

Author Response: We have revised and corrected the reference style.

Concern16: Check ref [14].

Author Response: We have revised and corrected the reference style.

Concern 17: Check ref [26].

Author Response: We have revised and corrected the reference style.

Concern 18: Check ref [33].

Author Response: We have revised and corrected the reference style.

Concern 19: Check ref [42].

Author Response: We have revised and corrected the reference style.

 Concern 20: Check all references one more time especially with co-authors first and last names, and volume, issue, and page numbers.

 Author Response: We have revised and corrected the issue of reference style.

Reviewer 3 Report

For Li-ion battery in electric vehicle (EV) applications, the authors proposed a single LC tank base cell-to-cell active voltage balancing algorithm. The authors described that the benefits of the proposed technique are fast balancing speed and high-power efficiency. However, the reviewer cannot understand the novelty of this work. The topology of the proposed voltage equalizer is well-known and have already been proposed in the following papers:

Uno and K. Yoshino, "Modular Equalization System Using Dual Phase-Shift-Controlled Capacitively-Isolated Dual Active Bridge Converters to Equalize Cells and Modules in Series-Connected Lithium-Ion Batteries," in IEEE Transactions on Power Electronics, doi: 10.1109/TPEL.2020.3013653. (See Fig. 3 (c))

As the authors of the above-mentioned articles explained, the proposed topology of this paper has a major drawback of the increased section switch count. Besides, due to the lack of comparison data with the state-of-the-art articles, the novelty of the proposed technique is not clear.

Reviewer’s other comments:

  1. As the reviewer pointed above, the novelty of the proposed topology is not clear. The proposed topology has already been proposed by other researchers. In the point of “novelty”, the reviewer cannot accept this paper.

  1. The authors must quote the references according to the reference number. Where are Refs. [14, 15]? The authors must check your manuscript before submission.

  1. The problem definition of this work is not clear. In Sect.1, the drawbacks of each conventional technique should be described clearly. The quotation of existing techniques is so rough, such as “The cell-to-cell (C2C) [26-44] balancing circuit could be charged and discharged the weak cells and strong cells efficiently. In C2C balancing circuit energy transfer from higher cell to lower cell by single capacitor, inductor, transformer [26-29], multi capacitor, inductor, transformer [30-32], cuk converter [33-34], boost converter [35-36], buck-boost converter [34,37-39], ramp converter [40], resonant converter [41-42], flyback converter [43-46]”. The authors must discuss the existing techniques one by one and should emphasize the difference with other methods to clarify the position of this work further.

  1. In the Introduction part, strong points of this proposed method should be further stated. The authors only described that “In this developing circuit, we reduce the number of bi-directional MOSFET switches and associate components”. The new features of the proposed method and the main advantages of the results over others should be clearly described.

  1. The authors should improve the figure presentation. Some of the figures are not clear. Please use clear images.

  1. Please unify the font style. In sentences/equations, mathematical expressions must be Italic font. (Some of them are Italic fonts and others are Roman font.) Otherwise, readers will be confused.

  1. In the theoretical analysis shown in Fig. 2, the authors must describe the theoretical assumption. Though some of analysis are presented, this paper is lack of rigorously theoretic derivation. Show the theoretic derivation clearly.

  1. In Sect. 4, the simulation conditions are not clear. Please explain the physical model of circuit components, such as transistor switches. As you may know, the simulated result strongly depends on the physical model of circuit components.

  1. Fig. 10 must be improved. Nobody can understand the gridline in Fig. 10. Besides, there is no XY-axis label. Readers will not be able to understand the meaning of these pictures.

  1. Due to thin comparison data, the effectiveness of the proposed technique is not clear. The authors compared the proposed technique with Refs. [27, 34, 39, 30]. However, these are old techniques proposed more than 5 years ago. Besides, the results shown in Fig. 13 are not supported by any mathematical reasons. The authors should justify the effectiveness of the proposed method by comparing with the latest methods. Even if “Energies (MDPI)”, the similar papers have already been proposed:

Hannan, M.A.; Hoque, M.M.; Ker, P.J.; Begum, R.A.; Mohamed, A. Charge Equalization Controller Algorithm for Series-Connected Lithium-Ion Battery Storage Systems: Modeling and Applications. Energies 2017, 10, 1390.

  1. In Sect. 5, there is no comparison data about power efficiency, power consumption, and response speed. To clarify the effectiveness of the proposed technique, the authors must demonstrate and compare theses characteristics. Otherwise, readers will not be able to understand the effectiveness of the proposed technique.

  1. The reviewer cannot understand the summary shown in Table 2. The discussion shown in Table 2 is not supported by any demonstrations. Besides, some of the data are suspicious. Please show the quantitative data.

  1. The results of this research are not clear in Conclusions. Furthermore, the benefits of the proposed method are not supported by theory. So, I fail to understand the scientific contribution of this research.

  1. The results of this research are not clear in Conclusions. Show the scientific contribution of this work with concrete data.

Author Response

Reviewer 3: For Li-ion battery in electric vehicle (EV) applications, the authors proposed a single LC tank base cell-to-cell active voltage balancing algorithm. The authors described that the benefits of the proposed technique are fast balancing speed and high-power efficiency. However, the reviewer cannot understand the novelty of this work. The topology of the proposed voltage equalizer is well-known and have already been proposed in the following papers:

Uno and K. Yoshino, "Modular Equalization System Using Dual Phase-Shift-Controlled Capacitively-Isolated Dual Active Bridge Converters to Equalize Cells and Modules in Series-Connected Lithium-Ion Batteries," in IEEE Transactions on Power Electronics, doi: 10.1109/TPEL.2020.3013653. (See Fig. 3 (c))

As the authors of the above-mentioned articles explained, the proposed topology of this paper has a major drawback of the increased section switch count. Besides, due to the lack of comparison data

Concern 1: As the reviewer pointed above, the novelty of the proposed topology is not clear. The proposed topology has already been proposed by other researchers. In the point of “novelty”, the reviewer cannot accept this paper.

Author Response:  

 This proposed cell-to-cell balancing circuit is bidirectional, where Uno and Yoshino proposed converter was unidirectional. Our proposed circuit reduces the balancing time and increases the balancing efficiency (94.8%) where Uno and Yoshino proposed circuit balancing time is too high and low efficiency ( average 84%). This proposed circuit works on three different modes: charging, discharging, and relax mode, and can be implemented for module-to-module balancing.  

Concern 2: The authors must quote the references according to the reference number. Where are Refs. [14, 15]? The authors must check your manuscript before submission.

 Author Response: We have corrected by citing references no 14 and 15 in the main text.

Concern 3:

The problem definition of this work is not clear. In Sect.1, the drawbacks of each conventional technique should be described clearly. The quotation of existing techniques is so rough, such as “The cell-to-cell (C2C) [26-44] balancing circuit could be charged and discharged the weak cells and strong cells efficiently. In C2C balancing circuit energy transfer from higher cell to lower cell by single capacitor, inductor, transformer [26-29], multi capacitor, inductor, transformer [30-32], cuk converter [33-34], boost converter [35-36], buck-boost converter [34,37-39], ramp converter [40], resonant converter [41-42], flyback converter [43-46]”. The authors must discuss the existing techniques one by one and should emphasize the difference with other methods to clarify the position of this work further.

Author Response: In this paragraph, we have discussed the cell-to-cell balancing system and. based on your advice we added the discussion among all of the conventional C2C balancing techniques and the issues in the Appendix. 

Discussion of all balancing circuits are:

Single capacitor, inductor, and transformer-based balancing circuits are balanced the cell voltage between two cells in a string. Here, 2n bidirectional switches or 2n + 2n diodes are used with a single capacitor, inductor, or transformer. These balancing circuits are faster and more efficient than passive balancing. However, they have required a complex control system, ripple current, and magnetizing loss. Multi capacitor, inductor, and transformer-based balancing circuits are balanced the cell voltage between two adjoined cells in the string. Here, 2n single switches are used for n cells. These balancing circuits have good balancing speed, medium complexity. However, they have high ripple current, magnetizing, and core loss.

Cuk converter is consists of two switches, a capacitor, and two inductors that are applicable for two cells. The charge balancing speed is high but the control system is complicated and faces magnetizing loss. In a boost, the converter inductor is used in parallel that can transfer the energy from any cell to any cell in the string. It can transfer the energy from an adjoined cell or any cell but this circuit faces magnetizing loss and ripple current. Buck-boost converter is the combination of buck and boost converter and works like a multi inductor-based balancing circuit. This circuit has good energy efficiency but required smart voltage sensing and a complex control system.

Ramp converter is an upgrade version of the multi-winding transformer-based balancing system. In this converter, the transformer secondary winding connected with cells, and the primary winding is connected with a single cell through of voltage detection circuit. This balancing circuit has a complex control system and hardware implication. The resonant converter is one of the most suitable charge balancing circuit. This circuit can transfer the energy from higher cell to lower cell between adjoined or any cell to any cell. This balancing circuit has a high balancing speed, efficiency, and low power loss. However, it required a complex control system. The flyback converter is also used in the cell balancing system in high power applications. It is also used in P2P, P2C, C2P balancing system. It has a high balancing speed and efficient but required smart voltage sensing, intelligent control, and face core magnetizing loss.

Concern 4:

In the Introduction part, strong points of this proposed method should be further stated. The authors only described that “In this developing circuit, we reduce the number of bi-directional MOSFET switches and associate components”. The new features of the proposed method and the main advantages of the results over others should be clearly described.

Author Response:  We have discussed our findings systematically. When the reader reads the resonant converter (in reference, 41-42) they can understand the difference between the proposed circuit and the existing circuit. In section 4, in the simulation result, we discussed how we chose the switching frequency. Finally, in conclusion, we explained the reason for the LC chosen. That is “To design the resonant tank large inductor and capacitor have been used because the resonant current wave is larger than the small inductor and capacitor resonant current wave. From this, the circuit loss will decrease and the balancing circuit takes short balancing time”. We updated the manuscript by highlighting the main contributions at the end of the introduction section, page number 2. “

Concern 5:

The authors should improve the figure presentation. Some of the figures are not clear. Please use clear images.

Author Response:  We have improved figure regulations.

Concern 6:

Please unify the font style. In sentences/equations, mathematical expressions must be Italic font. (Some of them are Italic fonts and others are Roman font.) Otherwise, readers will be confused.

  Author Response: We have updated all equations in Italic font also added a list for defining the nomenclature and abbreviations section on page 1.

Concern 7:

In the theoretical analysis shown in Fig. 2, the authors must describe the theoretical assumption. Though some of analysis are presented, this paper is lack of rigorously theoretic derivation. Show the theoretic derivation clearly.

Author Response: We have added more analysis based on the reviewer's comments. The analysis areas: “The direct C2C single balancing circuits are bidirectional, work on battery cells charging, relaxation mode and discharge period. All of the circuits are low voltage and current stress, good efficiency, miniature size, low cost, most suitable for low power application and can be used in P2P or module-to-module balancing system. However, single C2C balancing circuits are required complex control systems, sometimes face ripple current.”   

Concern 8:  In Sect. 4, the simulation conditions are not clear. Please explain the physical model of circuit components, such as transistor switches. As you may know, the simulated result strongly depends on the physical model of circuit components.

 Author Response: For the simulation, we used MATLAB SIMULINK 2016a software. Based on your comment we add this line for a clear explanation “To execute the simulation, we used ideal component from Simulink library”.

Concern 9:   Fig. 10 must be improved. Nobody can understand the gridline in Fig. 10. Besides, there is no XY-axis label. Readers will not be able to understand the meaning of these pictures.

Author Response: We have improved the figure by adding the XY-axis label.

Concern 10: Due to thin comparison data, the effectiveness of the proposed technique is not clear. The authors compared the proposed technique with Refs. [27, 34, 39, 30]. However, these are old techniques proposed more than 5 years ago. Besides, the results shown in Fig. 13 are not supported by any mathematical reasons. The authors should justify the effectiveness of the proposed method by comparing with the latest methods. Even if “Energies (MDPI)”, the similar papers have already been proposed:

Hannan, M.A.; Hoque, M.M.; Ker, P.J.; Begum, R.A.; Mohamed, A. Charge Equalization Controller Algorithm for Series-Connected Lithium-Ion Battery Storage Systems: Modeling and Applications. Energies 2017, 10, 1390.

Author Response: In the comparison, the above-mentioned references [27, 34, 39, 30] were our typing mistakes, now we fixed the appropriate Refs that’s are [29, 36, 41, 42]. Where we chose the most relevant reference [41] and [42] that are presented resonant converter and the reader can easily understand the improvement of our proposed circuits. In Figure 13, we did the comparison based on balancing the result between the proposed circuit with a single transformer-based balancing result [29], Quasi-resonant based balancing circuit [36], and resonant converter [41 & 42].

  Concern 11: In Sect. 5, there is no comparison data about power efficiency, power consumption, and response speed. To clarify the effectiveness of the proposed technique, the authors must demonstrate and compare theses characteristics. Otherwise, readers will not be able to understand the effectiveness of the proposed technique.

 Author Response: In Sect. 5.4 we have discussed the comparison among the converter based balancing and the key point of comparison presented in Table 2.  Based on your recommendation we added balancing speed, power loss, and control system in Table 2.  

Concern 12: The reviewer cannot understand the summary shown in Table 2. The discussion shown in Table 2 is not supported by any demonstrations. Besides, some of the data are suspicious. Please show the quantitative data.

Author Response: The comparison shown in Table 2 is provided from the rigorous study of existing research articles cited in references. Especially we focused on the Refs. [2, 15, 41, 42]

Concern 13: The results of this research are not clear in Conclusions. Furthermore, the benefits of the proposed method are not supported by theory. So, I fail to understand the scientific contribution of this research.

 Author Response: We have revised the conclusion section highlighting the findings and the results.

“To design, the resonant tank large inductor and capacitor have been used because the resonant current wave is larger than the small inductor and capacitor resonant current wave. From this, circuit loss will decrease and the balancing circuit takes short balancing time (for two 4200 mAh, 3.7 V Li-ion cells it takes 76 minutes)”

Concern 14: The results of this research are not clear in Conclusions. Show the scientific contribution of this work with concrete data.

 Author Response: We have revised the conclusion by highlighting the results with supportive data.

Round 2

Reviewer 1 Report

Fig 2,5,6,7,8,10,11,12 are not clear again. please improve quality of them.

Author Response

Concern 1: Fig 2,5,6,7,8,10,11,12 are not clear again. please improve quality of them.

Author Response: We have improved the picture quality based on your suggestion.

Reviewer 3 Report

In the revised version, some of the reviewer’s comments were improved well. However, due to the lack of concrete data, the reviewer cannot understand the effectiveness of the proposed converter. The authors must perform the quantitative comparison by demonstrating concrete data. In this paper, the authors’ interpretation is not supported by any demonstration. Please improve the comparison and discussion of this paper.

-----

Author Response for the 1st comment:

This proposed cell-to-cell balancing circuit is bidirectional, where Uno and Yoshino proposed

converter was unidirectional. Our proposed circuit reduces the balancing time and increases the

balancing efficiency (94.8%) where Uno and Yoshino proposed circuit balancing time is too high

and low efficiency (average 84%). This proposed circuit works on three different modes: charging,

discharging, and relax mode, and can be implemented for module-to-module balancing.

> Reviewer’s 1st comment:

> As the authors of the above-mentioned articles explained, the proposed topology of this paper has a major drawback of the increased section switch count.

-----

Reviewer’s comment for the revised version:

As Ueno’s paper pointed out, “the proposed topology of this paper has a major drawback of the increased section switch count.” The reviewer cannot understand why the size of the proposed converter is “small”. (see Table 2) There is no quantitative data in Table 2. Please demonstrate the size of each converter concretely. Besides, the authors described that “Our proposed circuit reduces the balancing time and increases the balancing efficiency (94.8%) where Uno and Yoshino proposed circuit balancing time is too high and low efficiency (average 84%).” However, this is not an apple-to-apples comparison. The authors must compare these converters under the same conditions.

-----

-----

Author Response for the 11th comment: In Sect. 5.4 we have discussed the comparison among the converter based balancing and the key point of comparison presented in Table 2. Based on your recommendation we added balancing speed, power loss, and control system in Table 2.

> Reviewer’s 11th comment:

>In Sect. 5, there is no comparison data about power efficiency, power consumption, and response speed. To clarify the effectiveness of the proposed technique, the authors must demonstrate and compare theses characteristics. Otherwise, readers will not be able to understand the effectiveness of the proposed technique.

-----

Reviewer’s comment for the revised version:

As Table 2 shows, there is no quantitative data in Table 2. The authors' opinion shown in Table 2 is not supported by any concrete data. Please demonstrate quantitative data to support the authors’ interpretation. As pointed out above, the reviewer disagrees with the authors’ opinion shown in Table 2.

-----

-----

> Reviewer’s 11th comment: The comparison shown in Table 2 is provided from the rigorous study of existing research articles cited in references. Especially we focused on the Refs. [2, 15, 41, 42]

> Reviewer’s 12th comment: The reviewer cannot understand the summary shown in Table 2. The discussion shown in Table 2 is not supported by any demonstrations. Besides, some of the data are suspicious. Please show the quantitative data.

-----

Reviewer’s comment for the revised version:

Due to the lack of proof, the reviewer disagrees with the authors’ opinion. Please demonstrate quantitative data to support the authors’ interpretation.

-----

Author Response

Reviewer 3: In the revised version, some of the reviewer’s comments were improved well. However, due to the lack of concrete data, the reviewer cannot understand the effectiveness of the proposed converter. The authors must perform a quantitative comparison by demonstrating concrete data. In this paper, the authors’ interpretation is not supported by any demonstration. Please improve the comparison and discussion of this paper.

Author Responses: We add the specific reference number with every balancing topology in Table 2 So that reader easily understands the balancing topology.

Author Response for the 1st comment:

This proposed cell-to-cell balancing circuit is bidirectional, where Uno and Yoshino proposed converter was unidirectional. Our proposed circuit reduces the balancing time and increases the balancing efficiency (94.8%) where Uno and Yoshino proposed circuit balancing time is too high and low efficiency (average 84%). This proposed circuit works on three different modes: charging, discharging, and relax mode, and can be implemented for module-to-module balancing.

> Reviewer’s 1st comment:

> As the authors of the above-mentioned articles explained, the proposed topology of this paper has a major drawback of the increased section switch count.

Author responses: Mr.Ueno’s present their proposed topology major drawback of the increased section switch count but we compare our work and ueno’s proposed circuit for the cell to cell balancing system.

Reviewer’s comment for the revised version:

As Ueno’s paper pointed out, “the proposed topology of this paper has a major drawback of the increased section switch count.” The reviewer cannot understand why the size of the proposed converter is “small”. (see Table 2) There is no quantitative data in Table 2. Please demonstrate the size of each converter concretely. Besides, the authors described that “Our proposed circuit reduces the balancing time and increases the balancing efficiency (94.8%) where Uno and Yoshino proposed circuit balancing time is too high and low efficiency (average 84%).” However, this is not an apple-to-apples comparison. The authors must compare these converters under the same conditions.

Author responses: Mr.Ueno’s present Modular Equalization System whereas our proposed circuit specially focused on the cell Equalization. This circuit is small in size because we reduce the number of switches, diodes, and associate components of LC energy carriers compare with [41, 42] that added in 367 & 268 lines.

Author Response for the 11th comment: In Sect. 5.4 we have discussed the comparison among the converter based balancing and the key point of comparison presented in Table 2. Based on your recommendation we added balancing speed, power loss, and control system in Table 2.

> Reviewer’s 11th comment:

>In Sect. 5, there is no comparison data about power efficiency, power consumption, and response speed. To clarify the effectiveness of the proposed technique, the authors must demonstrate and compare these characteristics. Otherwise, readers will not be able to understand the effectiveness of the proposed technique.

Author responses: In Sect. 5, comparison data is presented for battery cell equalization/balancing based on our rigorous study. Power efficiency, power consumption, and response speed are not applicable to battery cell equalization/balancing. These are applicable for microgrid, smart grid, vehicles to grid, or grid to vehicles system.

Reviewer’s comment for the revised version:

As Table 2 shows, there is no quantitative data in Table 2. The authors' opinion shown in Table 2 is not supported by any concrete data. Please demonstrate quantitative data to support the authors’ interpretation. As pointed out above, the reviewer disagrees with the authors’ opinion shown in Table 2.

Author Responses: We add the specific reference number with every balancing topology in Table 2 So that reader easily understands the balancing topology.

> Reviewer’s 11th comment: The comparison shown in Table 2 is provided from the rigorous study of existing research articles cited in references. Especially we focused on the Refs. [2, 15, 41, 42]

> Reviewer’s 12th comment: The reviewer cannot understand the summary shown in Table 2. The discussion shown in Table 2 is not supported by any demonstrations. Besides, some of the data are suspicious. Please show the quantitative data.

Author Responses: We presented the quantitative data in Table 3.

Reviewer’s comment for the revised version:

Due to the lack of proof, the reviewer disagrees with the authors’ opinion. Please demonstrate quantitative data to support the authors’ interpretation.

Author Responses: We presented the quantitative data in Table 3.

Round 3

Reviewer 3 Report

Reviewer’s comments:

1.

>Author Response for the 11th comment:

>In Sect. 5.4 we have discussed the comparison among the converter based balancing and the key point of comparison presented in Table 2. Based on your recommendation we added balancing speed, power loss, and control system in Table 2.

---

Concerning balancing speed only, the authors added the quantitative data in the revised version. However, other characteristics, such as power loss, efficiency, size and cost, were not demonstrated in Tables 2 and 3. As the reviewer pointed out, the authors’ interpretation is not supported by any quantitative data. Please show the quantitative data in Table 2. Readers will not be able to understand it. The authors should delete it if the authors cannot prove it with scientific data. Obviously, this is overemphasis.

2.

What’s “?” in Table 1? The authors should indicate it. (see 13.)

Author Response

Reviewer 3:

  1. Concerning balancing speed only, the authors added the quantitative data in the revised version. However, other characteristics, such as power loss, efficiency, size and cost, were not demonstrated in Tables 2 and 3. As the reviewer pointed out, the authors’ interpretation is not supported by any quantitative data. Please show the quantitative data in Table 2. Readers will not be able to understand it. The authors should delete it if the authors cannot prove it with scientific data. Obviously, this is overemphasis.

 Author Responses: The authors sincerely thank the reviewer for his/her great comment. We do agree with the reviewer to some extent, therefore we have included the quantitative data of power loss and efficiency in Table 2 and removed the voltage/current stress, speed, and cost comparison according to recommendations.

  1. What’s “?” in Table 1? The authors should indicate it. (see 13.)

Author Responses: The authors sincerely apologize for making such inconveniences. The paper has been revised and placed the “part name” instead of “?”

Round 4

Reviewer 3 Report

In this paper, the authors proposed a single LC tank base cell-to-cell active voltage balancing algorithm for Li-ion batteries in electric vehicle (EV) applications. In the first version, due to the lack of quantitative comparison data, the effectiveness and advantage of the proposed technique were not clear. However, in the revised version, the weak point was significantly improved. The revised version is well written and organized paper, I think. It is scientifically sound and contains sufficient interest to merit publication. This is a good paper.